# Diversity of Endomycorrhizal Fungi in Argan Forest Stands: Implications for the Success of Reforestation Programs

**Matike Ganoudi** [1,*], **Imane Ouallal** [2], **Abdelali El Mekkaoui** [3], **Majid Mounir** [4], **Mohammed Ibriz** [5] and **Driss Iraqi** [1]

1 Biotechnology Research Unit, Center of Agricultural Research of Rabat National Institute of Agricultural Research, Avenue Ennasr, BP 415 Rabat Principal, Rabat 10090, Morocco; driss.iraqi@inra.ma

2 Laboratory of Natural Resources and Sustainable Development, Ibn Tofail University, Kenitra 14000, Morocco; imaneouallal@gmail.com

3 Environment and Conservation of Natural Resources, Research Unit, Regional Center of Agricultural Research of Rabat, National Institute of Agricultural Research, Avenue Ennasr, BP 415 Rabat Principal, Rabat 10090, Morocco; abdelali.elmekkaoui@inra.ma

4 Biotechnology and Biotransformations Laboratory, Food Science and Nutrition Department, Hassan II Institute of Agronomy and Veterinary Medicine, P.O. Box 6202, Rabat 10112, Morocco; m.mounir@iav.ac.ma

5 Department of Biology, Genetics and Biometrics Laboratory, Faculty of Sciences, BP 133, Kenitra 14000, Morocco; mohammed.ibriz@uit.ac.ma

* Correspondence: matike.ganoudi@inra.ma; Tel.: +212-621123831

**Abstract:** Over the last few decades, argan trees (*Argania spinosa* L.) skeels have faced harsh ecological conditions and anthropogenic pressure, leading to a dramatic decline in surface and density of cultivation. Nowadays, most techniques used to regenerate argan trees have failed. Arbuscular mycorrhizal fungi (AMF) are root symbionts that increase plant resistance to biotic and abiotic stresses during transplantation. The exploration of these symbiotic fungi from different soils of argan stands is the starting point for the selection and production of high-performance organisms adapted to the reforestation sites. The objective of this study is to investigate the composition of the AMF community associated with the argan tree rhizosphere. Forty adult argan trees were sampled in eight forest sites representative of the distribution and genetic diversity of argan forest stands. Five sub-samples of rhizospheric soil were taken around each tree. Our results revealed the presence of different AMF structures (i.e., hyphae, vesicles/and arbuscules) in root samples. Based on morphological characterization, six genera of AMF spores were identified with a dominance of the genera *Septoglomus* (34%). In addition, soil organic matter and phosphorus concentrations showed a highly significant correlation with AMF spore density. The chi-square test showed a highly significant dependence of the distribution of genera on the site conditions of forest stands. These AMF could be tested and used during the inoculation of argan seedlings in forest nurseries for the success of restoration and reforestation programs, as well as for the development and sustainable improvement of this agroforestry system.

**Keywords:** *Argania spinosa*; rhizosphere; arbuscular mycorrhizal fungi (AMF); diversity; reforestation

## 1. Introduction

The argan tree is an endemic tree of Morocco belonging to the Sapotaceae family. It is a thermo-xerophilic species that grows in the center-west of Morocco between Agadir and Essaouira, Argan stands occupy an area of 871,210 hectares [1], and it is ranked as the second forest tree species in this country. The argan tree is considered a valuable species because it provides several useful parts. The wood is used as fuel (mainly as charcoal). The argan cake is the seed waste produced during the extraction process, and it is traditionally used for skin care and livestock nutrition. Different biological activities of argan cake have been cited, essentially as antioxidant, chemoprotective, anti-inflammatory, and antimicrobial [2]. Argan

oil is highly used for the manufacture of cosmetic products such as oil, soap, shampoo, and cosmetic creams, as well as for human consumption [3]. On the other hand, argan forests protect the soil against erosion, prevent desertification, and preserve ecological balance and biodiversity [4]. However, drought and strong anthropogenic pressure hamper the natural regeneration of the species and limit its range. To this end, several artificial regeneration programs have been implemented to improve and rehabilitate the degraded areas of the argan tree. To remedy this problem, it is necessary to improve the techniques of plant production in vitro and in nurseries. The results of ecological surveys carried out in the Sousse Massa region have shown that the argan tree, similar to the majority of forest species, forms endomycorrhizal associations. It was also noted during these surveys a predominance and wide distribution of the genera *Glomus* [5]. Arbuscular mycorrhizal fungi (AMF) have received a great deal of interest over the past twenty years, due to their favorable effects, mainly on the uptake of water and nutrients, particularly the weakly mobile phosphorus ion from the soil [6]. Inoculation of plants with AMF can not only facilitate their establishment but also improve the physicochemical and biological properties of the soil [7–9]. Argan reforestation requires young tree seedlings to be raised in nurseries and quickly adapt to the dry climate of the native range of these trees. Mycorrhizal inoculation significantly increases the growth and health of young argan seedlings, thereby increasing their fitness and survival after planting [10,11].

According to previous results [12], a local variant of an AMF species is more beneficial than a foreign species for its host. Therefore, it is essential to explore autochthonous AMF to select high-performance symbionts adapted to reforestation sites. Hence, the search for diversified, potent fungi with a broad spectrum of applications in reforestation programs is crucial. In this context, the objective of this work is to determine the diversity of communities of AMF associated with the argan tree in different edapho-climatic conditions in the northeast, northwest, and center-west of Morocco.

## 2. Materials and Methods

### 2.1. Sampling Sites in Argan Ecosystems

Eight sites of argan trees located in the northeast region (Aklim/Benisnassen), northwest region (Rabat/Oued Cheratt), and center-west region (Agadir, Ait Melloul, Essaouira, Ouled Taima, Talouine, and Taroudant), were prospected. These stations represent different eco-geographical and climatic conditions in Moroccan argan ecosystems (Figure 1).

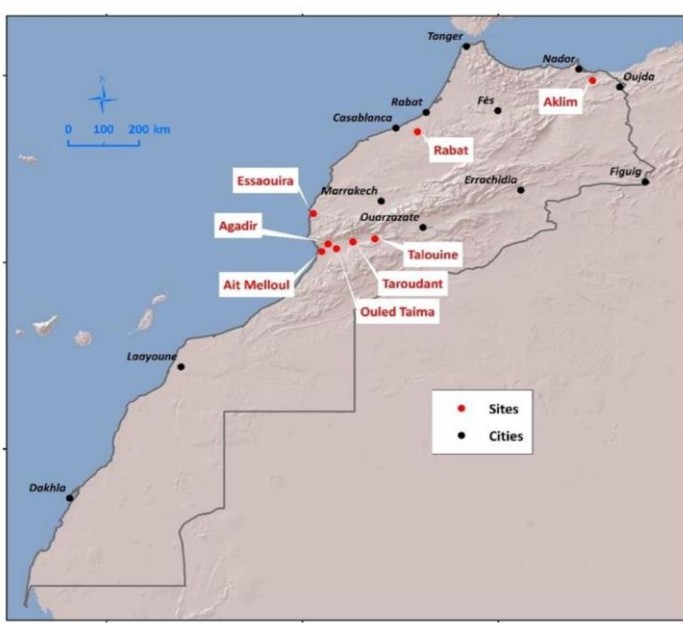

**Figure 1.** Location of the sampling sites.

## 2.2. Collection of Soil and Root Samples

Samples were collected from the eight argan stands in February, during the peak of the rainy season, when AMF populations often reach their maximum sporulation (Kaushal, 2000). First, we removed the roots growing beside the argan trees. Argan roots were distinctive for their hardness and pale red color.

Forty adult argan trees, representative of the general appearance of the tree at the eight stations, were sampled. To collect samples, five sub-samples of rhizospheric environments were taken around each tree at depths between 15 and 30 cm, where the finest roots likely to be mycorrhizal can be found. Care was taken not to damage the rootlets of the argan trees that were harvested. The five samples of "rhizospheric soils" from each argan tree were mixed in a plastic container to obtain a composite average sample (2 kg) that was representative of the rhizosphere of the sampled tree. Thus, five composite samples were taken at each station.

At the same time, samples of fine roots, likely to be mycorrhizal and more easily observable under a microscope, were collected from each argan tree in a bottle containing a solution of glycerol, ethanol, and distilled water (1 v/1 v/1 v). All samples were numbered, darkened, and transferred to the laboratory, where they were stored in a refrigerator at 4 °C for further analyses.

## 2.3. Physical and Chemical Analyses of Sampled Soil

The main physicochemical properties of the soil samples were analyzed according to the following conventional methods: The properties measured were pH in water at 1:1, and available phosphorus was determined by a colorimetric method as detailed in Olsen and Somers [13]. Total nitrogen was determined using the Kjeldahl method, and organic matter was determined using the Walkley and Black method [14,15]. This method consists of oxidizing the organic matter in the soil with a mixture of potassium dichromate in a hot sulfuric medium until the release of $CO_2$. The analysis of sodium and potassium was determined by photometry according to the extraction method adapted for ammonium acetate.

## 2.4. AMF Root Colonization

Root colonization was estimated following the staining method described by Phillips and Hayman [16]. Approximately 3 g of fine roots, cut into 0.5–1.0 cm pieces, were first cleared in 10% KOH and placed in a water bath at 90 °C for 1 h. They were rinsed with deionized water, neutralized with 1% HCl for a few minutes, and bleached with a solution of $H_2O_2$ for 10 min at 90 °C in the water bath. Finally, they were stained with 2% ink (Parker blue ink, USA) in 1% HCl at 70 °C for 30 min. The roots were subsequently cut into 1 cm lengths and mounted on slides for microscopic observations.

The evaluation of mycorrhizal colonization is performed on batches of 30 stained root fragments and mounted in three groups of ten between a slide and coverslip in lactic acid. The samples are then carefully examined under a microscope at a magnification of 100× to note the mycorrhizal structures. Occasionally, a magnification of 400× is used to better distinguish between the arbuscular and vesicular structures. The parameters noted include the frequency (the number of mycorrhizal root fragments) and intensity (proportion of colonized cortex estimated in relation to the entire root system) percentages of mycorrhization of the root system, as well as the contents of arbuscules and vesicles manifested inside the root cortex. These parameters are determined according to the method of (Trouvelot et al. [17].

## 2.5. Extraction of AMF Spores from Rhizosphere Soil Mixtures and Identification of Spores

The "wet sieving and decanting" method described by Gerdemann and Nicolson [18] and adapted by Cranenbrouck et al. [19] was used to isolate the spores of AMF. About 100 g of composite representative soil from each site was submerged in 1 L of water in a beaker and stirred for 1 min with a spatula. The supernatant was passed through four superimposed decreasing meshes sieves (500, 250, 106, and 50 μm). This operation was

repeated twice. This combination is suitable for collecting AMF spores. The upper sieve (500 μm) retains soil and root debris as well as mycorrhizal roots, while the other sieves (250, 106, and 38 μm) retain spores belonging to all the AMF genera. The contents of the 80 and 50 μm sieves are recovered with water (approximately 40 mL) in two beakers, which are then transferred into four tubes and centrifuged for 5 min at 5000 rpm. The supernatant, containing light debris and dead spores, is quickly removed, and the remaining volume (about 20 mL) is kept for the next step. The soil material was resuspended in a 1.4 M sucrose solution and centrifuged at 1000 rpm for 4 min. The spores are washed very carefully with water through a fine sieve (50 μm) to remove the remaining sucrose solution. These spores are then collected in a small Petri dish (50 mm) for observation under a microscope. The estimate of the number of spores in each soil is made by counting the spores in 1 mL of supernatant and extrapolating the total volume to 100 mL. The spores are collected one by one. The fungal spores recovered were first presented under a binocular magnifying glass (×40), then under a microscope (×100). Spores were considered viable if they had clear contents and an intact wall. They have been classified according mainly to their morphological characteristics (size, color, shape, consistency, wall structure, attachment of the suspensory hypha and ornamentation, and lipid reserves) based on the INVAM website (http://invam.caf.wvu.edu, accessed on 20 June 2022).

### 2.5.1. Spores' Density

Spore density was determined by direct counting under a binocular loupe. The total number of spores present in 100 g of soil was calculated as the average of five samples.

### 2.5.2. Species Richness

Genus richness was determined by calculating the number of morphotypes observed per site sample.

### 2.5.3. Occurrence Rate

The distribution of AMF genera between the studied sites was calculated by the percentage of sites where each genus was detected (the percentage of sites including a particular genus).

### 2.5.4. Shannon and Pielou Index

The Shannon index [20] expresses diversity by taking into account the number of genera and the abundance of individuals within each of these genera. It is calculated as follows:

$$H' = -\sum_{i=1}^{s} pi \ \log_2(pi)$$

pi = proportional abundance or percent importance of genera: pi = ni/N; S = Total number of genera
ni = number of individuals of genera in the sample
N = total number of individuals of all genera in the sample.
The Pielou index [21], which is the regularity of the distribution of species, is calculated as follows:
E = H'/$\log_2$ (S)
S = Total number of genera
H' = Shannon index

### 2.6. Statistical Analyses

A one-way ANOVA followed by a Tukey honestly significant difference post-hoc test ($p \leq 0.05$) was used to test the arcsine-transformed percentages of argan root colonization. The normal distribution of residuals was checked before analysis. A Pearson correlation was conducted at $p \leq 0.05$ to assess the relationships between mycorrhizal colonization and available P. A factor analysis of correspondence was performed on the AMF community composition by sites of sampling using the XLSTAT software.

All statistical analyses were performed using the SAS software database.

## 3. Results

### 3.1. Soil Physical Parameters

The results of the physicochemical properties of argan soil at the eight studied sites (Table 1) showed that the Taliouin and Ouled Taima soils are rich in clay (32% and 22%, respectively), while the Aklim soil s are rich in silt (75%), and the Ait Melloul soils are rich in sand (79%). The pH of rhizospheric samples taken from the various argan tree sites is alkaline, ranging from 7.2 for the Rabat site to 8.2 for the Taroudant site. Available phosphorus in the sampled soils is consistently low at all sites, and potassium content reaches 1971.2 ppm in the Aklim soil. Soils from Aklim and Agadir have the highest organic matter and total carbon contents.

**Table 1.** Physical and chemical properties of soil samples.

| Sites | % Clay | % Silt | % Sand | pH Water | pH KCl | P₂O₅ ppm | K₂O ppm | % SOM | % SOC |
|---|---|---|---|---|---|---|---|---|---|
| Agadir | 20.6 ± 0.4 d | 36.4 ± 0.2 d | 43.6 ± 0.3 d | 7.5 ± 0.1 bc | 7.2 ± 0.1 b | 21.1 ± 15.9 c | 588.8 ± 0.4 b | 4.3 ± 0.1 a | 1.8 ± 0.0 b |
| Ait Melloul | 4.9 ± 0.1 f | 15.3 ± 0.2 h | 78.9 ± 0.1 a | 7.3 ± 0.2 c | 7.3 ± 0.1 ab | 47.2 ± 0.0 b | 360.2 ± 0.1 f | 2.1 ± 0.1 bc | 1.3 ± 0.0 d |
| Aklim | 5.7 ± 0.1 f | 74.8 ± 0.2 a | 19.2 ± 0.2 g | 7.6 ± 0.1 bc | 7.5 ± 0.a | 14.1 ± 0.1 cd | 1971.2 ± 0.1 a | 4.4 ± 0.1 a | 2 ± 0.1 b |
| Essaouira | 9.0 ± 0.1 e | 44.7 ± 0.4 b | 44.4 ± 0.4 d | 7.8 ± 0.0 ab | 7.1 ± 0.1 b | 4.2 ± 0.0 d | 161.5 ± 0.1 g | 4.2 ± 0.1 a | 0.8 ± 0.0 e |
| Ouled Taima | 21.6 ± 0.2 c | 29.5 ± 0.3 | 48.1 ± 0.1 c | 7.4 ± 0.1 bc | 7.2 ± 0.1 b | 30.4 ± 0.1 cb | 404.1 ± 0.1 e | 0.9 ± 0.0 d | 0.6 ± 0.0 f |
| Rabat | 20.1 ± 0.2 d | 24.7 ± 0.1 f | 36.8 ± 0.1 e | 7.2 ± 0.1 bc | 7.3 ± 0.1 ab | 29.1 ± 0.1 c | 134.4 ± 0.1 h | 1.9 ± 0.1 c | 2.8 ± 0.0 a |
| Taliouin | 32 ± 0.5 a | 43.1 ± 0.5 c | 24.4 ± 0.3 f | 7.7 ± 0.1 bc | 7.2 ± 0.1 b | 69.0 ± 0.1 a | 555.7 ± 0.1 c | 2.03 ± 0.1 c | 1.2 ± 0.0 d |
| Taroudant | 20.9 ± 0.2 b | 23.2 ± 0.2 g | 53.8 ± 0.5 b | 8.2 ± 0.0 a | 7.1 ± 0.1 b | 47.2 ± 0.1 b | 495.1 ± 0.1 d | 2.3 ± 0.0 b | 1.3 ± 0.1 c |

Mean values within the same column followed by different letters differ significantly ($p < 0.05$) using Tukey's test HSD at 5%. Data are presented as mean ± standard deviation ($n = 3$). SOM: soil organic matter, SOC: soil organic carbon.

### 3.2. Root Colonization and Spore Density of AMF

Roots from eight sites were examined, and different structures were observed, including arbuscules, vesicles, and extracellular hyphae (Figure 2), in all roots from the sites examined. The results show a significant difference between sites. The frequency of mycorrhization was high at all the sites studied, with values ranging from 80% to 100%. Similarly, mycorrhization intensity varied significantly between sites, with the highest level observed in Aklim (37% ± 6.27) and the lowest level in the Taroudant site (20% ± 4.01). The percentage of vesicle colonization was significantly higher at the Taliouin site (53% ± 10.97) compared to other sampling sites. Conversely, no significant difference in arbuscular colonization was observed between sites (Table 2). Examination of the rhizospheric soils of *Argania spinosa* showed that the average spore density varied from site to site, with the highest at the Essaouira site (120 spores per 100 g of soil) and the lowest at the Taroudant site (40 spores per 100 g of soil) (Table 2).

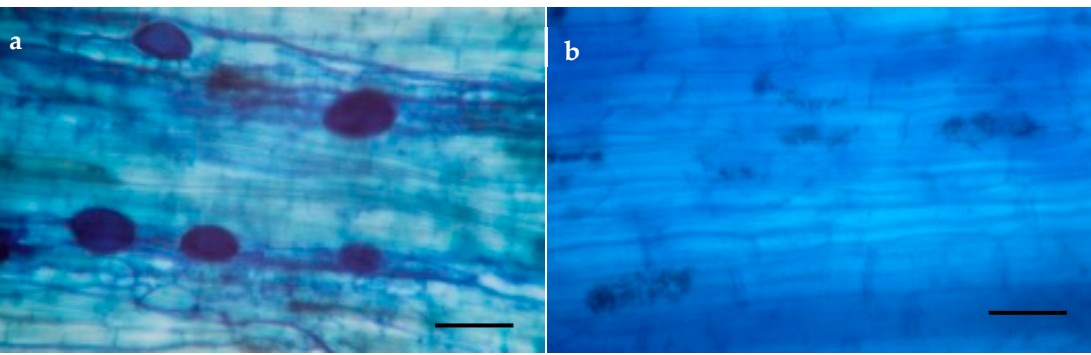

**Figure 2.** The different structures observed in the roots collected from the argan tree: (**a**) vesicles and hyphae; (**b**) Arbuscules. 1 cm = 100 μm.

**Table 2.** Mycorrhizal Frequency, intensity, arbuscular, and vesicular contents of argan roots.

| Sites | % F | % I | % V | % A | SD/100 g of Soil |
|---|---|---|---|---|---|
| Agadir | 100.0 ± 0.0 a | 22.3 ± 2.3 b | 35.3 ± 4.2 ab | 52.2 ± 6.4 a | 101 a |
| Ait Melloul | 72.0 ± 3.0 a | 26.1 ± 4.5 ab | 42.7 ± 4.0 a | 50.2 ± 9.9 a | 60 b |
| Aklim | 98.0 ± 1.6 ab | 36.6 ± 6.2 a | 34.5 ± 3.8 abc | 52.0 ± 5.7 a | 98 ab |
| Essaouira | 99.6 ± 0.3 bc | 30.8 ± 2.7 a | 31.5 ± 2.7 abc | 51.8 ± 6.1 a | 120 a |
| Ouled Taima | 84.0 ± 5.5 c | 25.7 ± 2.4 ab | 41.1 ± 2.9 a | 49.8 ± 3.6 a | 57 bc |
| Rabat | 84.0 ± 3.2 c | 22.9 ± 2.0 b | 21.5 ± 3.7 bcd | 52.4 ± 1.6 a | 46 bc |
| Taliouin | 80.0 ± 4.4 cd | 27.3 ± 10.3 ab | 52.8 ± 10.9 a | 38.2 ± 12.2 a | 50 bc |
| Taroudant | 88.0 ± 6.0 d | 19.5 ± 4.0 b | 48.7 ± 7.6 a | 40.3 ± 6.3 a | 40 c |

Mean values in the same column followed by different letters differ significantly ($p < 0.05$) using the Tukey HSD test at 5%, data presented as mean ± standard deviation ($n$ = 5). %F: mycorrhizal frequency, %I: mycorrhizal intensity, %V: vesicular, %A: arbuscular, and SD: spore density.

### 3.3. AMF Community Composition in Soils

Our research revealed that the isolated spores belonged to the three family glomale orders, Glomeraceae, Gigasporaceae, and Acaulosporaceae. The analysis of the morpho-anatomical characteristics of this spore community revealed the presence of six genera: Septoglumus, Acaulospora, Rhizophagus, Glomus, Funneloformis, and Racocetra (Figure 3).

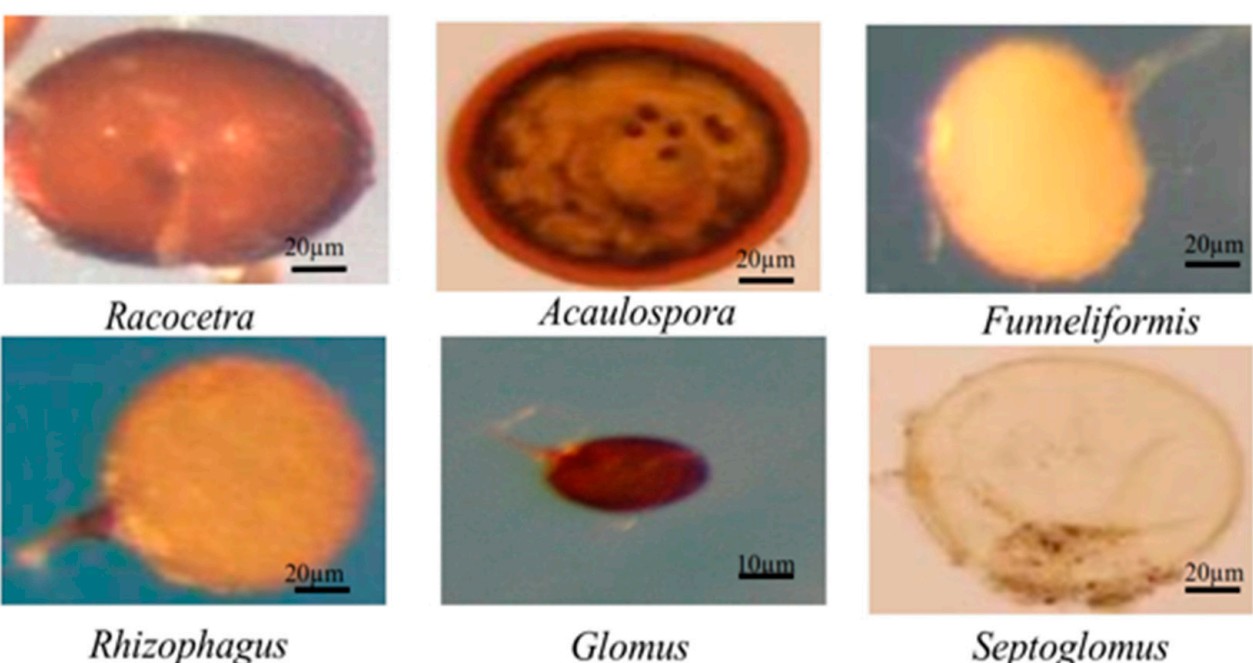

**Figure 3.** Types of arbuscular mycorrhizal fungi isolated from the argan tree rhizosphere.

The analysis of the relative abundance of AMF genera presents in soil samples from different sites revealed the presence of only one genus at Ait Melloul (Septoglomus) and Aklim (Rhizophagus) sites, the presence of two genera at Agadir and Taroudant (Septoglomus, Rhizophagus), Rabat (Rhizophagus, Racocetra), and Taliouin (Septoglomus, Funneliformis) sites, and the presence of three genera (Septoglomus, Glomus, Acaulospora) at Essaouira site. Regarding the occurrence rate of the genera, Septoglomus was present in 34% of the surveyed sites, followed by Rhizophagus in 22%, while the other genera were present in 11% of the studied sites (Figure 4).

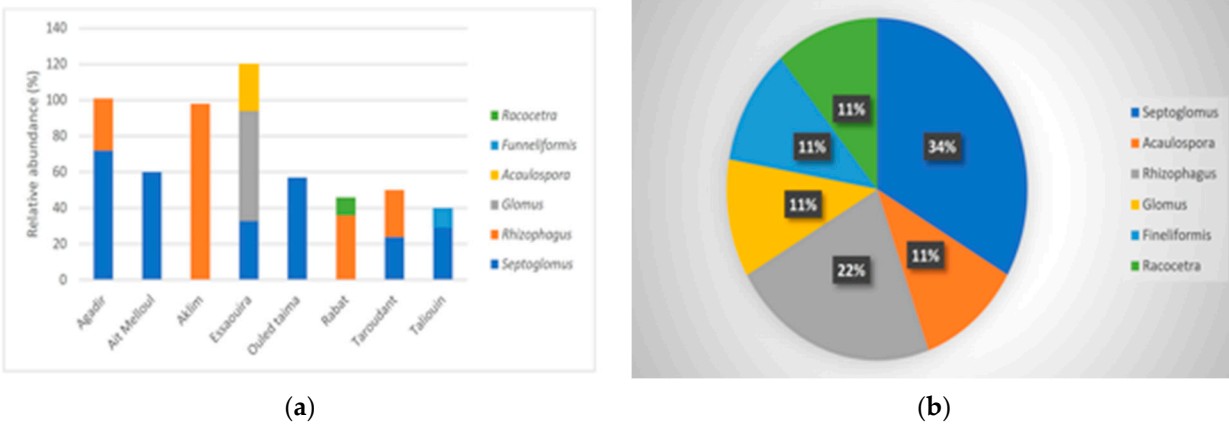

(**a**)                                                                          (**b**)

**Figure 4.** (**a**) Relative abundance of arbuscular mycorrhizal fungi genera presents in soil samples from different sites, (**b**) occurrence rate of each genus found under the argan tree rhizosphere.

Factorial analysis of correspondences (FAC) showed a highly significant correlation between the sites and genera of AMF ($p < 0.0001$). Furthermore, it subdivided the sites into two classes: Class 1 (Taliouin, Oueled Taima, Ait Melloul, Agadir, Aklim, Taroudant, and Rabat) and Class 2 (Essaouira) (Figure 5).

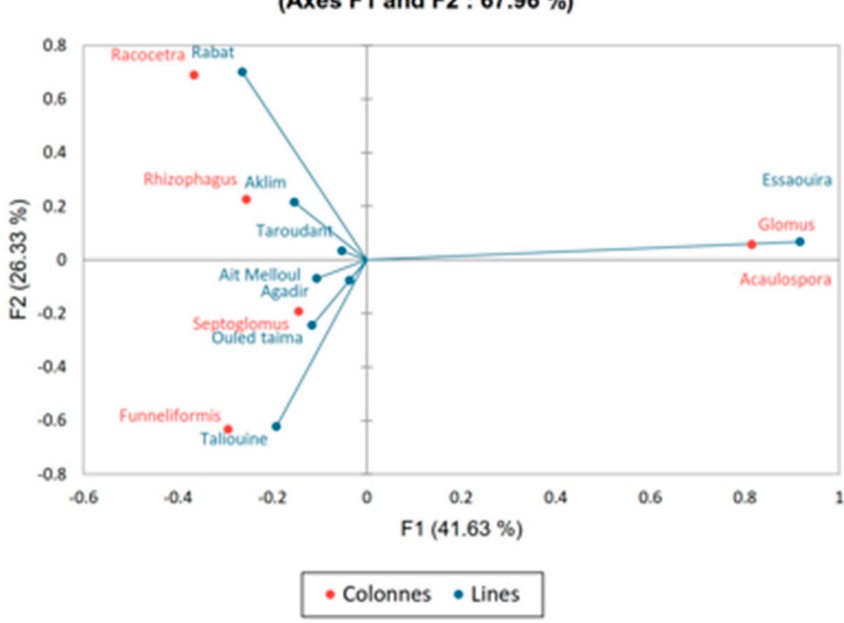

**Figure 5.** Factorial analysis of correspondences (FAC) of arbuscular mycorrhizal fungi community composition according to sampling sites.

The relationship between physicochemical soil properties and AMF spore density is represented in Table 3. In fact, a positive and highly significant correlation was observed between spore density (SD) and soil organic matter ($r = 0.8802$, $p \leq 0.01$). On the other hand, a highly significant inverse correlation was observed between spore density and $P_2O_5$ ($r = -0.8688$, $p \leq 0.01$).

**Table 3.** Pearson correlation coefficients between soil physicochemical parameters and arbuscular mycorrhizal fungi spore density (SD) of argan rhizosphere soil samples.

| | %Clay | %Silt | %Sand | pH Water | pH KCl | P$_2$O$_5$ ppm | K$_2$O ppm | %SOM | %SOC |
|---|---|---|---|---|---|---|---|---|---|
| SD | −0.58754 | 0.54890 | −0.12613 | −0.05867 | −0.10489 | −0.86885 | 0.29033 | 0.88022 | −0.11987 |
| | 0.1256 | 0.1589 | 0.7660 | 0.8902 | 0.8048 | 0.0051 ** | 0.4855 | 0.0039 ** | 0.7774 |

Pearson Correlation Coefficient, N = 8 Prob > | r | under H0: Rho = 0; SOM: Soil Organic Matter, SOC: Soil Organic Carbon. Effects marked with ** are significant (** $p \leq 0.01$).

Calculation of the Shannon diversity index showed a specific diversity with different abundances between the sites studied. Indeed, the Taliouin and Ouled Taima sites showed the highest diversity index compared to the other sites. Similarly, both sites had the highest Pielou index (E). The Pielou index (E) showed a highly variable distribution of genera between sites, indicating an imbalance in the AMF community present in the eight sites studied (Figure 6).

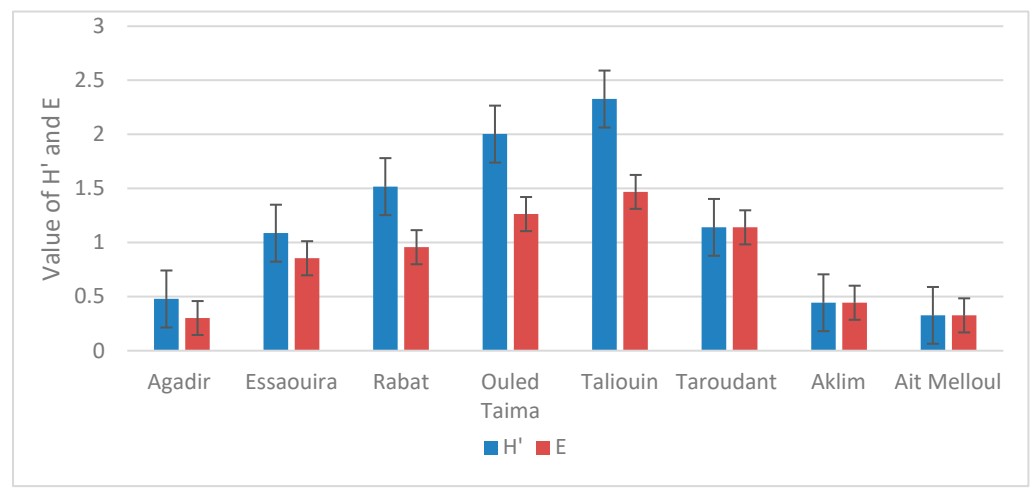

**Figure 6.** Shannon index (H') and the Pielou (E) index of the eight stations, standard deviation (*n* = 5).

## 4. Discussion

Knowledge of AMF diversity in forest areas is an important biological parameter for the assessment of environmental disturbances. The use of indigenous soil mycorrhizae has advantages in quick acclimatization and ecosystem restoration, such as their good tolerance to local conditions and low ecological risk [22]. Indeed, the study will provide a basis for AMF utilization in the success of argan reforestation programs in the region.

According to studies conducted in the surveyed areas, the roots of the argan tree carry endomycorrhizal structures: vesicles, arbuscular, and hyphae. This is explained by several authors who confirm the dependence of this tree on AMF [22–24]. The quantified mycorrhization frequencies were always high (80%–100%) in all the argan plantations studied, which confirms that *Argania spinosa* is a highly mycotrophic species. The highest mycorrhizal frequencies were noted at the sites of Agadir (100%), Essaouira (99.6%), and Aklim (98%). The mycorrhizal intensity varied between 36% for the Aklim site and 19% for the Taroudant site. In addition, our study showed a relationship between the number of spores and the mycorrhizal intensity, which was confirmed by some authors [25,26]. According to [27,28], the sporulation may depend on the type of AMF, soil characteristics, and climatic conditions. The contents of arbuscules and vesicles change from one site to another.

The variability of the parameters of root colonization by AMF may be explained by differences in the physicochemical properties of the soils and the climatological conditions of the sites studied. Root colonization variability is strongly dependent on plant fitness,

and in some situations, a reduced root biomass owing to dryness might result in a higher frequency of AMF colonization [29].

The results of the available phosphorus content of the soil samples collected from the different sites were low. Indeed, the negative correlation recorded between the levels of root colonization (Frequency and intensity) and the concentration of available phosphorus confirmed the adaptation of AMF to poor soils in P [30–32]. These results are also consistent with those reported by several authors [10,24], who state that the frequencies of mycorrhization are high in soils with low levels of total phosphorus in argan. However, this relationship does not seem to be valid on other sites.

The results of AMF isolation from the argan rhizosphere of the studied sites showed variations in the abundance and number of morphotypes. The highest density was observed in the Essaouira plain site (120 spores per 100 g of soil). This number is much lower than that reported by several authors: 207 spores per 100 g [24], 188 spores per 100 g [10], and 1128 spores per 100 g [33]. Red. [34] reported 900 à 2080 spores per 100 g, which is also higher than our finding. These results also remain low compared to those found in the rhizosphere of certain Moroccan species, such as *Tetraclinis articulata* [35], *Cupressus atlantica* [36], *Olea europaea* [37], *Date palm* [38]. The highest spore densities (120, 101, and 98 spores per 100 g of dry soil) were observed in the sites of Essaouira, Agadir, and Aklim, respectively, which had high organic matter content (4.2, 4.3, and 4.4%). Indeed, it has been noted that the addition of organic amendments (manure, compost, and crop residues) can stimulate the multiplication of AMF [39]. Furthermore, the differences recorded may be due to the physicochemical and microbiological properties of soils in arid and semi-arid areas [40], microclimate fluctuations [41], vegetation cover, sampling season [42], spore formation, and deterioration of germination [43].

The preliminary identification, based only on the morphological criteria of the spores, showed the presence of six genera of AMF (*Septoglumus*, *Acaulospora*, *Rhizophagus*, *Glomus*, *Funneloformis*, and *Racocetra*) belonging to three families (*Glomaceae, Gigasporaceae,* and *Acaulosporaceae*). These results are very important compared to those of Ouallal et al. [24], who reported only two genera (*Glomus and Scutellospora*) of arbuscular mycorrhizal fungi in the nine Moroccan argan trees that they studied. In addition, Elmaati et al. [33] revealed the dominance of the *Rhisophagus* and *Glomus* genera in five argan sites. However, Sellal et al. [10] identified five genera (*Glomus*, *Scutellospora*, *Entrophospora*, *Pacispora*, and *Gigaspora*) from spores isolated in the rhizosphere of argan trees from seven sites in the regions of Essaouira, Agadir, Taroudant, and Tiznit.

The results of the occurrence rate revealed the dominance of the *Septoglomus* genera in the sampling sites, with a distribution rate of 34%, followed by *Rhizophagus* with 22%, while the other genera were present in 11% of the studied sites. The dominance of the genera *Septoglomus* in several different environments indicates high plasticity and high adaptation to different impacts of biotic or abiotic origin [44]. According to the authors, the dominance of *Glomus* is due to its ability to produce more spores in less time than other genera, such as *Gigaspora* and *Acaulospora*. This dominance of the *Glomus* genera has also been reported in several studies on the argan tree rhizosphere [10,24,33].

According to Chagnon et al. [45], *Acaulospora* is considered to be stress tolerant. In fact, it has frequently been reported in harsh climatic circumstances, acidic soils (i.e., pH 3.6–4.20; Morton, 2017), and high-altitude locations, at 2500 m [46]. In our investigation, this genus was found in alkaline soils (pH 7.81) at 150 m above sea level. This shows the capacity of the genus to thrive in a variety of different environments. *Acaulospora* generally predominates in dry forests but not in wet forests [47].

In general, *Glomus*, *Rhizophagus*, *Septoglomus*, *Funneliformis*, and *Acaulospora* were already identified in the rhizosphere of the argan tree in semi-arid areas.

The Shannon Diversity and Pielou index indicated that the AMF community was highly diverse with respect to abundance and species distribution; this study was consistent with results found by Ouallal et al. [24], who showed a similar diversity between study sites. The indices used here are among the most commonly used in AMF studies [48].

## 5. Conclusions

To our knowledge, our results report for the first time the composition of the arbuscular mycorrhizal fungi community in several forest sites representative of the genetic diversity and ecological distribution of argan forest stands: the northeast region (Aklim/Bnisnassen), the northwest (Rabat/Oued cheratt), and the center-west (Agadir, Ait Melloul, Essaouira, Ouled Taima, Taliouin, and Taroudant). The results revealed a strong correlation between the density of AMF spores and the physicochemical properties of the soils. The results also showed an abundance and a diversification of AMF associated with the species. In all sites, we identified six genera of AMF (*Septoglomus*, *Rhizophagus*, *Glomus*, *Acaulospora*, *Fineloformus*, and *Racocetra*). This richness highlighted in the rhizosphere of the argan tree is very important; it gives useful information for the selection of an effective inoculum according to the different pedoclimatic zones. This diversity encourages further research into the selection of native AMF species that show good adaptation to environmental conditions and climatic variation between sites. It is well established that AMF provides vigorous and robust plants that withstand transplanting and weather conditions. Indeed, the mycorrhization of plants in Moroccan nurseries with native inoculum before their transfer to the field is an essential step for the success of plantations in reforestation programs and the restoration of degraded ecosystems.

**Author Contributions:** Conceptualization, M.G. and D.I.; methodology, M.G.; software M.G.; validation, D.I.; formal analysis, M.G. and I.O.; investigation, M.G.; resources, M.G.; data curation M.G. and I.O.; writing—original draft preparation M.G.; writing—review and editing, I.O.; visualization, M.G.; supervision, D.I.; project administration M.G., I.O., A.E.M., M.M., M.I. and D.I.; funding acquisition, D.I. All authors have read and agreed to the published version of the manuscript.

**Funding:** This research and the publication fees were supported by National Institute of Agronomic Research Avenue Ennasr, BP 415 Rabat Principal, Rabat 10090, Morocco.

**Data Availability Statement:** Publicly available sources of the data used in this study are described in the article. However, other relevant data or information can be requested from the authors on reasonable grounds.

**Acknowledgments:** This study was carried out as part of the INRA research project. We thank the technicians and researchers of the Biotechnology Laboratory for their valuable advice. We would also like to thank the Soil Analysis Laboratory of the Environment and Conservation of Natural Resources Research Unit, INRA, in Rabat, for their help in analyzing the soil sample, and Fatima Gaboun (INRA) for her support with statistical analysis.

**Conflicts of Interest:** The authors declare no conflict of interest.

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
