# Peer review of "Diversity of Endomycorrhizal Fungi in Argan Forest Stands: Implications for the Success of Reforestation Programs"

_forests, doi:10.3390/f14081649_

Round 1
Reviewer 1 Report
An important feature of arbuscular mycorrhizal fungi is the diversity of their community in natural ecosystems. The research progress in this area is quite rich, especially with the Next Generation Sequencing. From this starting point, the article does not show any innovation. Especially since similar research has been conducted in almost the same study sites (Ex. Sellal et al. 2016). The results obtained from the study are not very original.
Therefore, in my opinion, the article is not very innovative.
It is not known which reference database the authors used to identify the AMF spores.
The list of bibliographic references does not follow the rules for citing articles according to the journal guidelines. The name of some journals is not mentioned. Ex. Line 58: Specify the journal. Was this study published in 2002 or 2011? Please check carefully.
Therfore, due to the lack of innovation, the results that are not fully presented, a lack of illustrations with pictures of endomycorrhizal structures detected in argan roots but especially pictures of the different morphotypes and genera of AMF identified, I do not think the article should be published.
Author Response
It is not known which reference database the authors used to identify the AMF spores
Spores were induced as viable if they had clear contents with an intact wall. They have been classified according mainly to their morphological characteristics (size, color, shape, consistency, structure of the wall and attachment of the suspensory hypha) based on INVAM website (http://invam.caf.wvu.edu)
The list of bibliographic references does not follow the rules for citing articles according to the journal guidelines. The name of some journals is not mentioned. Ex. Line 58: Specify the journal. Was this study published in 2002 or 2011? Please check carefully.
It has been corrected
Therfore, due to the lack of innovation, the results that are not fully presented, a lack of illustrations with pictures of endomycorrhizal structures detected in argan roots but especially pictures of the different morphotypes and genera of AMF identified, I do not think the article should be published.
The figures have been added: page 2 and 6

Reviewer 2 Report
see an attachment

English proofreading is needed
Author Response
The english has been reviesed

Reviewer 3 Report
This study intends to assess the mycorrhizal status and diversity of AM fungi associated with argan trees growing in different sites in Morocco. The results of the study indicate that all the argan root samples examined possessed AM fungal colonization and spores of AM fungi belonging to six genera were found in the root zone soils. In addition, AM fungal spore numbers were correlated to soil organic matter and phosphorus. These results are not overwhelmingly exciting as there are some recent detailed studies where the diversity of AM fungi associated with argan trees was reported from Morocco. For instance, see Sellal et al. (2022, https://doi.org/10.5772/intechopen.106162). Further, the ability of indigenous AM fungi in promoting argan seedling growth under nursery conditions is also well demonstrated (Ouallal et al. 2018, Tree Planters' Notes 61: 35–44; Soufiani et al. 2022; https://doi.org/10.1080/11263504.2022.2048280). Therefore, the need for the present study is not clear. Moreover, the study is also confounded by several shortfalls as mentioned below.
1. The study involves some wrong methodologies. For instance, soil collected at a depth of 15–30 cm around a tree is not the rhizosphere soil. The soil region that is influenced by the roots (those attached to the roots after shaking the loose soil) is the rhizosphere soil. For more details, see https://doi.org/10.1007/s13213-012-0491-y.
2. Another important concern of the study is that the AM fungal spore enumeration and species diversity is based on soil samples collected from a single time point. It is well known that the sporulation of many AM fungi is seasonal and closely aligned to the phenology of the host plant. It is therefore important to collect soil samples during different time points in a year and during different years to bring out the actual diversity of AM fungi.
3. Likewise, the procedure adopted for extraction and enumeration of AM fungal spores from the soil samples fails to reveal the actual diversity of AM fungal spores in the soils. Non-examination of sievates from the >106µm sieve could have resulted in missing spores belonging to genera that usually produce larger spores (members of Gigasporaceae), and taxa that produce spores in clusters, or sporocarps. This along with sampling restricted to a one-time point of a year renders the AM fungal diversity assessment rather incomplete.
4. Yet another major shortfall of the study is that AM fungal diversity assessment based on field-collected spores has certain limitations. Generally, most of the field-collected spores are devoid of contents, parasitized by soil microorganisms, and exist as spore cases. Therefore, it is important to mention how the intact and healthy spores were distinguished from the spore cases, dead, and decaying spores.
5. The characters used in the identification of the AM fungal spores are not mentioned. For instance, some small-spored Gigasporaceae taxa without the subtending hyphae can resemble Acaulospora since both these taxa possess membranous walls. Therefore, it is important to specify the characters used to distinguish spores of different genera. Further, the number of spores in each type used for characterization is also missing.
6. Generally, a large number of AM fungal spores retrieved from the field soil are unidentifiable (even at the genus level) because of the lack of diagnostic characteristics. This is due to the influence of soil factors on spore morphology as previously mentioned or the developmental stages of spores. Moreover, spores of the same species may present different wall characters during various stages of development. It is therefore important to initiate trap cultures and later pure cultures to retrieve spores that could be identified with confidence. However, this was not attempted in the present study.
7. A wide range of endophytic fungi other than AM fungi colonizes plant roots growing in the field. Therefore, explain how the structures of AM fungi (especially hyphae) were distinguished from the non-mycorrhizal fungi.
8. Moreover, plants growing in natural vegetations have overlapping root zones (from conspecifics or different species) that grow in close proximity to the intended plant species. It is important to therefore clarify how this was taken care of in the present study.
9. The discussion is mostly comparative, weak, and empty. The discussion fails to adequately reason for the results observed.
10. Italicize the genus/scientific names throughout the manuscript. Similarly, use abbreviations once they are introduced. Certain abbreviations are introduced in several instances during the course of the presentation.
11. Tables should be self-explanatory. Therefore, avoid using abbreviations in the table titles.
12. The references listed should be thoroughly checked and corrected. There are incomplete references and scientific names in many instances are not italicized.
Other comments:
Line 32: The test is written as Chi-square and not Khi2.
Lines 36–37: Avoid repeating words that are already there in the title as keywords.
Line 41: Separate the words thermo-xerophilic species.
Line 44: Correct the punctuation mark.
Lines 55, 78, 104, 112: Separate the joined words. Do likewise for all related ones.
Line 57: Replace ‘endomycorrhizae’ with ‘endomycorrhizal’.
Lines 133 and 136: Express centrifugal force in g units.
Table 2: Separate the decimals with points and not commas.
Lines 264–265, 267–268, 277–278: Italicize the genus names.

The presentation must be improved as there are spelling errors, incomplete sentences, and wrong sentence structure throughout the manuscript.
Author Response
Point1. The study involves some wrong methodologies. For instance, soil collected at a depth of 15–30 cm around a tree is not the rhizosphere soil. The soil region that is influenced by the roots (those attached to the roots after shaking the loose soil) is the rhizosphere soil. For more details, see https://doi.org/10.1007/s13213-012-0491-y.
Response1. The sampling depth in our study is the same as that of the recommended article
Point 2. Another important concern of the study is that the AM fungal spore enumeration and species diversity is based on soil samples collected from a single time point. It is well known that the sporulation of many AM fungi is seasonal and closely aligned to the phenology of the host plant. It is therefore important to collect soil samples during different time points in a year and during different years to bring out the actual diversity of AM fungi.
Response 2. Smples were collected from the eight argan groves in February, during the peak of the rainy saison, when AMF populations often reach their maximum sporulation (Kaushal, 2000).
Point 3. Likewise, the procedure adopted for extraction and enumeration of AM fungal spores from the soil samples fails to reveal the actual diversity of AM fungal spores in the soils. Non-examination of sievates from the >106µm sieve could have resulted in missing spores belonging to genera that usually produce larger spores (members of Gigasporaceae), and taxa that produce spores in clusters, or sporocarps. This along with sampling restricted to a one-time point of a year renders the AM fungal diversity assessment rather incomplete.
Response 3. For the extraction of the samples, we used a series of sieves with decreasing superimposed mesh (500,250, 106, 38) (line 130- 131)
Point 4. Yet another major shortfall of the study is that AM fungal diversity assessment based on field-collected spores has certain limitations. Generally, most of the field-collected spores are devoid of contents, parasitized by soil microorganisms, and exist as spore cases. Therefore, it is important to mention how the intact and healthy spores were distinguished from the spore cases, dead, and decaying spores.
Response 4. Spores were induced as viable if they had clear contents with an intact wall
Point 5. The characters used in the identification of the AM fungal spores are not mentioned. For instance, some small-spored Gigasporaceae taxa without the subtending hyphae can resemble Acaulospora since both these taxa possess membranous walls. Therefore, it is important to specify the characters used to distinguish spores of different genera. Further, the number of spores in each type used for characterization is also missing.
Response 5. The AMF spores have been classified according mainly to their morphological characteristics (size, color, shape, consistency, structure of the wall and attachment of the suspensory hypha) based on INVAM website (http://invam.caf.wvu.edu). The attachment hypha and the walls really distinguish between Gigaspora and Acaulospora.
Point 6. Generally, a large number of AM fungal spores retrieved from the field soil are unidentifiable (even at the genus level) because of the lack of diagnostic characteristics. This is due to the influence of soil factors on spore morphology as previously mentioned or the developmental stages of spores. Moreover, spores of the same species may present different wall characters during various stages of development. It is therefore important to initiate trap cultures and later pure cultures to retrieve spores that could be identified with confidence. However, this was not attempted in the present study.
Response 6. Based on the morphological properties of the mycorrhizal spores, the taxa isolated from the argan rhizosphere were identified. These morphologically identified complexes then need trap culture to find strong sporulating genera, monoculture to gather spores that could be positively identified, and finally, molecular analysis to characterize and confirm AMF diversity. The majority of work carried out on the diversity of the AMF of the argan tree has identified species with a lack of diagnostic characteristics.
Point 7. A wide range of endophytic fungi other than AM fungi colonizes plant roots growing in the field. Therefore, explain how the structures of AM fungi (especially hyphae) were distinguished from the non-mycorrhizal fungi.
Response 7. AMF hyphae are often attached to arbuscules and vesicles
Point 8. Moreover, plants growing in natural vegetations have overlapping root zones (from conspecifics or different species) that grow in close proximity to the intended plant species. It is important to therefore clarify how this was taken care of in the present study.
Response 8. First, we removed the roots of plants growing beside the argan roots. Argan roots distinctive for their hardness and pale red color (line 84-85)
Point 9. The discussion is mostly comparative, weak, and empty. The discussion fails to adequately reason for the results observed.
Response 9. the discussion has been improved
Point 10. Italicize the genus/scientific names throughout the manuscript. Similarly, use abbreviations once they are introduced. Certain abbreviations are introduced in several instances during the course of the presentation.
Response 10. It has been corrected
Point 11. Tables should be self-explanatory. Therefore, avoid using abbreviations in the table titles.
Response 11. It has been corrected
Point 12. The references listed should be thoroughly checked and corrected. There are incomplete references and scientific names in many instances are not italicized.
Response 12. It has been corrected
Other comments:
Line 32: The test is written as Chi-square and not Khi2. It has been corrected
Lines 36–37: Avoid repeating words that are already there in the title as keywords.
Only the word diversity that is repeated, we can't remove it
Line 41: Separate the words thermo-xerophilic species. It has been corrected
Line 44: Correct the punctuation mark. It has been corrected
Lines 55, 78, 104, 112: Separate the joined words. Do likewise for all related ones. It has been corrected
Line 57: Replace ‘endomycorrhiza’ with ‘endomycorrhizal’. It has been corrected
Lines 133 and 136: Express centrifugal force in g units. It has been corrected
Table 2: Separate the decimals with points and not commas. It has been corrected
Lines 264–265, 267–268, 277–278: Italicize the genus names. It has been corrected

Round 2
Reviewer 3 Report
In this revised manuscript, the authors have taken into consideration some of the suggestions raised in the earlier version of the manuscript. However, there are still some major considerations that remain unattended as shown below.
Comment 1. The study involves some wrong methodologies. For instance, soil collected at a depth of 15–30 cm around a tree is not the rhizosphere soil. The soil region that is influenced by the roots (those attached to the roots after shaking the loose soil) is the rhizosphere soil. For more details, see https://doi.org/10.1007/s13213-012-0491-y.
Authors' Response: The sampling depth in our study is the same as that of the recommended article
My response: This is not correct as mentioned earlier not all the soils that are at the depth of 15–30 cm are rhizosphere soil. It is only the soil that is influenced by plant roots is the rhizosphere soil.
Comment 2. Another important concern of the study is that the AM fungal spore enumeration and species diversity is based on soil samples collected from a single time point. It is well known that the sporulation of many AM fungi is seasonal and closely aligned to the phenology of the host plant. It is therefore important to collect soil samples during different time points in a year and during different years to bring out the actual diversity of AM fungi.
Author's Response: Samples were collected from the eight argan groves in February, during the peak of the rainy season, when AMF populations often reach their maximum sporulation (Kaushal, 2000).
My response: Again, this is a wrong perception. During the rainy season, generally, plants produce new roots to satisfy the increased demand for plant growth. Therefore, spores that are present in the soil germinate to initiate colonization due to the presence of new roots. As AM fungi are obligate symbionts, sporulation happens at the end of the growing season when the plant and root growth slows down. Generally AM fungal sporulation does not occur at the peak plant growth phase as mentioned.
Comment 3: Likewise, the procedure adopted for extraction and enumeration of AM fungal spores from the soil samples fails to reveal the actual diversity of AM fungal spores in the soils. Non-examination of sievates from the >106µm sieve could have resulted in missing spores belonging to genera that usually produce larger spores (members of Gigasporaceae), and taxa that produce spores in clusters, or sporocarps. This along with sampling restricted to a one-time point of a year renders the AM fungal diversity assessment rather incomplete.
Author's Response: For the extraction of the samples, we used a series of sieves with decreasing superimposed mesh (500,250, 106, 38) (lines 130- 131).
My Response: It is again contents in sieves 106–38 µm that have been examined for AM fungal spores (Line 133) that excludes the large spore-producing species.
Comment 4: Yet another major shortfall of the study is that AM fungal diversity assessment based on field-collected spores has certain limitations. Generally, most of the field-collected spores are devoid of contents, parasitized by soil microorganisms, and exist as spore cases. Therefore, it is important to mention how the intact and healthy spores were distinguished from the spore cases, dead, and decaying spores.
Author's Response: Spores were induced as viable if they had clear contents with an intact wall.
My Response: Some of the images depicted in Figure 3 (Septoglomus and Acaulospora) are actually spore cases that are in the process of decomposition and not intact spores.
Comment 5: The characters used in the identification of the AM fungal spores are not mentioned. For instance, some small-spored Gigasporaceae taxa without the subtending hyphae can resemble Acaulospora since both these taxa possess membranous walls. Therefore, it is important to specify the characters used to distinguish spores of different genera. Further, the number of spores in each type used for characterization is also missing.
Author's Response: The AMF spores have been classified according mainly to their morphological characteristics (size, color, shape, consistency, structure of the wall, and attachment of the suspensory hypha) based on INVAM website (http://invam.caf.wvu.edu). The attachment hypha and the walls really distinguish between Gigaspora and Acaulospora.
My Response: Racocetra in Figure 3 lacks an apparent bulbous subtending hyphae characteristic of the taxa.
Comment 6: Generally, many AM fungal spores retrieved from the field soil are unidentifiable (even at the genus level) because of the lack of diagnostic characteristics. This is due to the influence of soil factors on spore morphology as previously mentioned or the developmental stages of spores. Moreover, spores of the same species may present different wall characters during various stages of development. It is therefore important to initiate trap cultures and later pure cultures to retrieve spores that can be identified confidently. However, this was not attempted in the present study.
Author's Response: Based on the morphological properties of the mycorrhizal spores, the taxa isolated from the argan rhizosphere were identified. These morphologically identified complexes then need trap culture to find strong sporulating genera, monoculture to gather spores that could be positively identified, and finally, molecular analysis to characterize and confirm AMF diversity. The majority of work carried out on the diversity of the AMF of the argan tree has identified species with a lack of diagnostic characteristics.
My Response: It is not clear how species identification could be performed using AM fungal spores that lacked diagnostic characters. This is also evident in the present study where identification appears based on intact spores (Figure 3) without any analysis of wall or subcellular characters. For instance, Septoglomus, Funneliformis, Rhizophagus, and Glomus were all under one genus (Glomus) and the spore morphology of these species tend to overlap. It is therefore important to mention how these closely related taxa were identified with confidence.
Comment 7: A wide range of endophytic fungi other than AM fungi colonizes plant roots growing in the field. Therefore, explain how the structures of AM fungi (especially hyphae) were distinguished from the non-mycorrhizal fungi.
Author's Response: AMF hyphae are often attached to arbuscules and vesicles.
My response: All AM fungal hyphae need not be attached to arbuscules or vesicles. In certain regions, the fungi can exist as only hyphae or hyphal coils. It is therefore important to specify the hyphal characters used in the delimitation of AM fungi.
The manuscript needs language editing to check the flow.
Round 2
Comment 1. The study involves some wrong methodologies. For instance, soil collected at a depth of 15–30 cm around a tree is not the rhizosphere soil. The soil region that is influenced by the roots (those attached to the roots after shaking the loose soil) is the rhizosphere soil. For more details, see https://doi.org/10.1007/s13213-012-0491-y.
Response 1. We absolutely agree with you, just the argan tree has a root system of pivot type with a dense network of superficial roots. For the argan tree the zone located at a depth between 15 -30 is influenced by the roots of the plant. According to the bibliography, it is the zone recommended by several authors who have worked on the diversity.
Comment 2. Another important concern of the study is that the AM fungal spore enumeration and species diversity is based on soil samples collected from a single time point. It is well known that the sporulation of many AM fungi is seasonal and closely aligned to the phenology of the host plant. It is therefore important to collect soil samples during different time points in a year and during different years to bring out the actual diversity of AM fungi.
Response 2. Our methodology of single-period sampling aims to highlight the difference between the mycorrhizal communities present in the different regions of the argan tree's presence (the north, center, east, and west). This gives the originality of our work a parallel to the published works in this direction (central-western Morocco).
Comment 3. Likewise, the procedure adopted for extraction and enumeration of AM fungal spores from the soil samples fails to reveal the actual diversity of AM fungal spores in the soils. Non-examination of sieves from the >106µm sieve could have resulted in missing spores belonging to genera that usually produce larger spores (members of Gigasporaceae), and taxa that produce spores in clusters, or sporocarps. This along with sampling restricted to a one-time point of a year renders the AM fungal diversity assessment rather incomplete.
Response 3. The spores from the 250-106-38 sieves were retained. it is corrected in the article line 132-133.
Comment 4. Yet another major shortfall of the study is that AM fungal diversity assessment based on field-collected spores has certain limitations. Generally, most of the field-collected spores are devoid of contents, parasitized by soil microorganisms, and exist as spore cases. Therefore, it is important to mention how the intact and healthy spores were distinguished from the spore cases, dead, and decaying spores.
Response 4. Due to the spore’s preservation up until the time of photo-taking.
Comment 5. The characters used in the identification of the AM fungal spores are not mentioned. For instance, some small-spored Gigasporaceae taxa without the subtending hyphae can resemble Acaulospora since both these taxa possess membranous walls. Therefore, it is important to specify the characters used to distinguish spores of different genera. Further, the number of spores in each type used for characterization is also missing.
Response 5. The photo of Racocetra has been changed in figure 3.
Comment 6. Generally, many AM fungal spores retrieved from the field soil are unidentifiable (even at the genus level) because of the lack of diagnostic characteristics. This is due to the influence of soil factors on spore morphology as previously mentioned or the developmental stages of spores. Moreover, spores of the same species may present different wall characters during various stages of development. It is therefore important to initiate trap cultures and later pure cultures to retrieve spores that can be identified confidently. However, this was not attempted in the present study.
Response 6. For the distinction between genera, we based ourselves on several morphological characters (size, color, shape, consistency, wall structure, attachment of the suspensory hyphae, ornamentation and lipid reserves) based on the INVAM site (http ://invam.caf.wvu.edu).
Comment 7. A wide range of endophytic fungi other than AM fungi colonizes plant roots growing in the field. Therefore, explain how the structures of AM fungi (especially hyphae) were distinguished from the non-mycorrhizal fungi.
Response 7. Intra-radicular hyphae are connected to arbuscules and extend extra-radicular into the soil to form a mycelial network.